# Application of Habitat Evaluation Procedure with Quantifying the Eco-Corridor in the Process of Environmental Impact Assessment

**DOI:** 10.3390/ijerph16081437

**Published:** 2019-04-23

**Authors:** Jiyoung Choi, Sangdon Lee

**Affiliations:** Department of Environmental Science and Engineering, Ewha Womans University, Seoul 03760, Korea; 01052201106@hanmail.net

**Keywords:** habitat evaluation procedure, ecological evaluation, road construction, roadkill, eco-corridor

## Abstract

In contrast to other fields, environmental protection (e.g., habitat protection) often fails to include quantitative evaluation as part of the existing environmental impact assessment (EIA) process, and therefore the EIA is often a poor forecasting tool, which makes selecting a reasonable plan of action difficult. In this study, we used the Habitat Evaluation Procedure (HEP) to quantify the long-term effects of a road construction project on an ecosystem. The water deer (*Hydropotes inermis*) was selected as the species of study since it uses an optimum habitat; water deer habitat data were collected on vegetation cover, stream water density, geographic contour, land use class, and road networks. The Habitat Suitability Index (HSI) and Cumulative Habitat Unit (CHU) values for the water deer were estimated to investigate the major land cover classes, the national river systems, and vegetation cover. Results showed that the environmental impact in the road construction project area would result in a net ecological loss value of 1211 without installation of an eco-corridor, which reduced to 662 with an eco-corridor, providing a 55% increase in the net value after 50 years of the mitigation plan. Comparing the 13 proposed ecological mitigation corridors, the corridor that would result in the highest net increase (with an increase of 69.5), was corridor #4, which was regarded as the most appropriate corridor to properly connect water deer habitat. In sum, the study derived the net increase in quantitative values corresponding with different mitigation methods over time for a road construction project; this procedure can be effectively utilized in the future to select the location of ecological corridors while considering the costs of constructing them.

## 1. Introduction

An environmental impact assessment (EIA) is often required to predict and assess the adverse effects of development projects, including a large-scale construction project, on the environment prior to their execution. However, in most cases in which an impact assessment is being carried out, typically the results are not accurate enough to predict impacts and ways to mitigate them. It has been shown that the current EIA system in Korea is limited in its ability to accurately estimate impacts on the natural environment and mitigation measures to reduce these impacts [1].

The current EIA framework provides a detailed assessment of the species of plants and animals and their distribution, and incorporates the concept of “Green Degree Naturality” and “Dominance and Sociability”, but lacks a comprehensive quantitative model to adequately protect the environment in the areas of development [1,2].

In addition, unlike some pollution measures that comprise a current EIA, the scope of measures for ecosystem protection are most likely to be determined by the developers themselves, making it more difficult to make accurate quantitative measurements of the environment as part of the EIA. As a result, there are many difficulties in assessing ecosystems and developing long-term monitoring systems for them.

The Habitat Evaluation Procedure (HEP) involves assessment based on accounting techniques in which the habitat of the target species can be quantified, and the ‘quality’ of the species’ habitat is the Habitat Suitability Index (HSI) value generated by calculating areas that conform to the target species’ habitat within its distribution. The HEP should be used as a method to develop mitigation measures. Usually, assessment of and mitigation of damage to the natural environment, including specific ecosystems, is rather difficult to quantify. The HEP can be used in the current EIA framework for selection of the project development sites, whereby site boundaries can be readjusted to promote evaluation of and mitigation measures for the natural environment [3,4]. Through HEP it should be possible to select which development alternatives are justified in terms of lowering habitat modification or damage. Large-scale projects often ignore the conservation of habitat areas that have been identified as having some form of particular importance, such that it often takes a long time for the habitat areas to recover from the damages incurred, which is an especially important consideration for habitats of endangered species [4].

The HEP can be applied to quantitatively predict the extent of damage to the natural environment via a model that compares the total habitat amount pre- and post-development. The HEP is the first habitat assessment model whose measures are designed to quantitatively identify the environmental value of an ecosystem in terms of its size and habitat quality and mirrors measures of the U.S. National Environmental Protection Act of 1969 by stipulating that (1) habitat can be expressed quantitatively for a specific species, and (2) habitat can be valued in terms of its quantity as well its quality. A certain amount of habitat of a certain quality should be designated for animal species since these variables directly influence their survival [4,5].

The purpose of this research was to measure the net ecological value of an eco-corridor before and after the road construction based on the HEP technique as a way to explore the method as an effective quantitative assessment supplement to the current EIA framework. The area chosen for assessment was that being set aside for the Hongcheon-Inje Expressway Construction Project. Our overall idea was to quantitatively evaluate and predict the impact on the area’s ecosystem from road construction by applying the relevant techniques of HEP in the EIA process.

## 2. Materials and Methods

### 2.1. Study Area

Of the total distance of 89.70 km between Chuncheon and Yangyang, the section between East Hongcheon and Inje is 36.85 km; the width of the road along this stretch is 23.4 m with the road consisting of four lanes. The highway construction was ecologically based, since 13 ecological routes were laid out in which the goal was to reduce or avoid roadkill, especially of water deer (*Hydropotes inermis*) (Figure 1). The time range for construction was 2006–2010, with a highway opening date of 2011 and an ecosystem target assessment of 2030 [6,7]. Eco-corridors were designed that were 40 m in width and 200–700 m in length, depending on the shape of mountain contour lines, and usually the top part was planted with vegetation of species found in surrounding areas in order to minimize the negative impact on the use of ecosystem structures by the local animals.

### 2.2. Study Species

The target species of our study was the water deer (*Hydropotes inermis*), which is a Korean endemic species. The species is internationally protected and was listed as vulnerable in 2008 by the International Union for Conservation Nature (IUCN) [8,9,10,11].

The water deer is the most common wild animal species in Korea, and water deer roadkill commonly appears along roads throughout Korea, including national roads and highways and smaller rural roads [12,13,14,15,16]. Based on data from the Korea Highway Corporation, from 2009 to 2013 the water deer was the most frequent roadkill species. With 84.5% (9078 individuals) of Korean roadkill attributed to the water deer, the species is the leader among the top nine roadkill species in the country (Appendix A
Table A1) [17].

Given the large amount of data on water deer roadkill, the species was selected for our analysis so that we could best evaluate the probability of roadkill numbers being mitigated by the creation of eco-corridors. It was assumed that the water deer would be the most frequently observed species during the field survey, as it is found in open, forest, and agricultural habitats.

### 2.3. Data Collection and Analysis

The area of study is the 36.85 km section between East Hongcheon and Inje. There are 13 ecological routes found within this section. The areas are known as a part of the Baekdudaegan mountain ranges, which is a continuous mountainous area in the Korean peninsula that supports the eastern part of Korea, with mountains featuring high peaks of more than 1000 meters.

Spatial analysis with a geographic information system (GIS) was used to coordinate and record the points where the highway construction ecological corridors were to be created. Water deer habitat data were collected covering vegetation cover, stream water density, contour lines, land use class, and road networks. The roadkill data were gathered by experts in the Korean Express Highway and they visited the highways in the early morning in order to identify roadkill.

To calculate the Habitat Suitability Index (HSI) and the Cumulative Habitat Unit (CHU), values for the water deer in the area of Hongcheon-Inje, the major land use classes [2], national river systems, and the vegetation cover along Highway No. 60 Hongcheon–Inje [6,7], which runs through the water deer distribution area, were investigated. Map data and information were extracted from the construction project’s report on the environmental impact assessment [7] and were entered into ArcView ver. 3.3 (Esri, Redlands, CA, USA), in which ArcGIS map ver. 9.3.1 (Esri, Redlands, CA, USA) and Spatial Extension for Arc View were used. The spatial analysis was performed using the stream networks, vegetation cover, land use classes, and road networks data, as well as the ecological and behavioral water deer data [18,19,20,21,22,23].

Map production and space analysis was performed with Arc View and GIS [24,25,26].

## 3. Theory/Calculations

### 3.1. Habitat Evaluation Procedure (HEP) Theory

The HEP employed an analytical method based on the use of a land coverage map to account for the habitat of the target species, and the ‘quality’ of the species’ habitat was generated to assess the value of habitat before and after the construction based on target species’ habitat within its distribution range (Table 1).


Suitability Index (SI)


The first stage of the study involved the collection of the environmental factor data (e.g., vegetation cover) for the water deer in order to generate a Suitability Index (SI), and the extraction of the environmental factor data in order to generate the variables. The environmental factors that were used to develop the HSI for the water deer in Korea included the variables that were analyzed by the US Geological Survey in order to generate the HSI for the white-tailed deer (*Odocoileus virginianus*) (Figure 2).


Distance from water (SI_1_)


Water deer prefer areas with water, so an Euclidean distance measure with spatial analyst tool was applied in GIS with a maximum distance of 1.6 km. The average value was determined to be 0.40 (s.e = ±0.03) with values over 0.6 for 35.57% of the area.


Distribution ratio of forest to feed (Food) (SI_2_)


Food resources of water deer using vegetation around the region was determined to be 0.76 (s.e = ±0.03), with values over 0.6 for 84.15% of the area. Mongolian oaks (*Quercus mongolica*) covered 47.84% of the area and mixed oak forests covered 32.44% of the area, indicating the existence of areas that were suitable habitat for water deer.


Density of shelter (Vegetation) (SI_3_)


Areas for vegetation density in the region were estimated to be 0.79 (s.e = ±0.03), with 58.63% of the area having the value of 1.0 and 39.20% of the area having medium value of 0.5, showing that most areas were suitable for water deer habitat.


Elevation (SI_4_)


Elevation was calculated with a contour map. Values were estimated 0.94 (s.e = ±0.05), with 81.31% of areas at an elevation of 400–800 m, indicating that most areas are presumably suitable for water deer.


Distribution ratio of the development areas (SI_5_)


A land coverage map indicated that 91.61% of the study area was comprised of forest areas and urban areas (0.98%), with an overall value was estimated to be 0.9.


Distance from road (SI_6_)


Using a similar method to SI_1_, the Euclidean distance was estimated. The average value was 0.69 (s.e = ±0.04), and most areas were 1.0 (50.57%) and greater than 0.6 (63.71%), indicating that areas were typically far from the road.

Based on the water deer HEP, habitat requirements were divided into the factors of water, food, shelter, and artificial interference. Selection of the SI was based on the existing literature, including expert studies on the species, and a habitat impact assessment methodology. After the extraction of the environmental factor data, six SIs were selected for water deer, in which the SI values of the species ranged from a minimum value of 0.0 to a maximum value of 1.0 (Figure 3).

### 3.2. Habitat Suitability Index (HSI)

After the SI data were extracted, the HSI values were obtained using the arithmetic mean method. After considering the relationship between the species and each of the six selected SI systems, the system deemed the most appropriate was fit by weighting the water deer data to the SI data.

This analysis is based on doubling the average distance of a water deer would need to travel to obtain its water (SI_1_) and food (SI_2_), which are essential environmental requirements of the species, and are the most important among other environmental factors [25,26].

For the assessment to function, all SI values should be zero, which in turn generates all zero HSI values. Importantly, the geometric mean method generally applies the arithmetic mean method, as discussed above, since the values using the latter tend to be smaller:(1)$$\mathrm{HSI}=\frac{2\left({\mathrm{SI}}_{1}+{\mathrm{SI}}_{2}\right)+{\mathrm{SI}}_{3}+{\mathrm{SI}}_{4}+{\mathrm{SI}}_{5}+{\mathrm{SI}}_{6}}{8}$$

### 3.3. Habitat Unit (HU) and Total Habitat Unit (THU) Assessment

The Habitat Unit (HU) value was generated with HSI values obtained for all study stages in each region multiplied by the HU values from each area of study. The cumulative HU value obtained for each study region is the Total Habitat Unit (THU) value. The HU evaluation plays an important role in both the present and future regional wildlife population planning [4,5,27].

The data from the 13 sites proposed for ecological corridors was introduced in the following formula:(2)$$\mathrm{THU}=\sum_{i=1}^{13}\mathrm{HU}i$$

The THU reflects a value at a given moment in time, and the THU values were generated for two target years (TYs) corresponding with the start and the completion of highway and eco-corridor construction.

Land classification of the eco-corridor areas was based on the environmental impact assessment conducted in the past year in which the land was classified based on its division into a radius of 1 km. The TYs for the THUs were the present year, and 3, 20, and 50 years.

The basis for the project’s 50 TY span is that this is the timeframe that was used for the HEP as per the U.S. National Environmental Protection Act (NEPA); for that HEP, the duration of the CHU analysis was for at least 50 years. Therefore, based on time and the mathematical function, the study assumed that the time needed for the restoration of the construction sites to their natural environmental states was 50 years.

### 3.4. Cumulative Habitat Unit (CHU) Assessment

The final step is to generate the CHU by integrating the previously obtained THU with the target time selected and calculating the graphical area of the portion surrounded by the time axis and the line graph indicating the variation in the THU. The value of the derived CHU minus the value of CHU from the graphical analysis—the case that there was no project—can be seen as the value of the net impact only, indicating that the ecological corridor, which was used as a mitigation measure, can be used [5,27].

## 4. Results

### 4.1. Habitat Evaluation Procedure (HEP) Assessment

#### 4.1.1. Habitat Suitability Index in Eco-Corridor Areas

The SI_1_ of the Hongcheon-Inje region averaged 0.40 with 35.57% of the area having an SI value greater than 0.6. Areas above the SI_2_ value of 0.6 accounted for 84.15% of the region, with an average value of 0.76. The SI_3_ averaged with 0.79 being a minimum value of 0.2 and a maximum value of 1.0. The SI_4_ averaged with 0.94 being a minimum value of 0.0 and a maximum value of 1.0 (Table 2, Table 3, Table 4, Table 5, Table 6 and Table 7).

The SI_5_ averaged 0.9, with 1.06% of the area able to be developed and where 91.61% of the area was forest. The SI_6_ average was 0.69 in terms of the road distance criterion, and 63.71% of the area could be developed; the SI_6_ had a minimum value of 0.6 and a maximum value of 1.0 (Table 7).

According to the HSI model, the fitness index for water deer from the Hongcheon-Inje region averaged 0.71. For regions where the HSI had a value of 0.6 or higher, the area was deemed as suitable for water deer habitat [27]; in terms of the percentage of areas with an SI value greater than 0.6, the areas are projected to decrease in the future from 63.7% to 55.9%.

#### 4.1.2. Total Habitat Unit Value Before and After Construction

The THUs calculated based on the HSI in consideration of environmental changes in time (TY) and the values of the area of the graph plotted as the CHU of the target year according to three different scenarios are shown in Figure 4.

The higher the final CHU value, the better the eco-corridor option associated with that value. The CHU value after highway construction was subtracted from the CHU value before construction, which gave the net loss value resulting from highway construction [12,13,14]. The value was calculated for a comparison with the eco-corridor effects in terms of habitat unit.

The CHU value for the study region before highway construction was 1211.8, and the CHU value after highway construction was 299.4; thus, the CHU net loss value was estimated as 912.4.

Moreover, the CHU value of 299.4 after highway construction and the value of 323 after the eco-corridor construction showed a net increase of 323. Thus, we conclude that the creation of a mitigating ecological corridor will act to increase the quality of water deer habitat.

### 4.2. Selection of Eco-Corridor

Among the 13 eco-corridors, the largest CHU net increase corresponded with option #4, which showed a CHU positive effect of 69.53, thus indicating that it is the most appropriate location for a habitat connection. The CHU values for each of the potential eco-corridor locations are shown in Table 8.

## 5. Discussion

Using the HEP, this study resulted in the selection of the most beneficial eco-corridor routes (in terms of the greatest net CHU value increases) among various eco-corridor routes in an area. Six factors were chosen for consideration, such as proximity to water, foods, vegetation density, elevation, ratio of the development areas, and distance to roads (Table 2, Table 3, Table 4, Table 5, Table 6 and Table 7). Based on the criterion of a Suitability Index (SI) value over 0.6, 63.7% of the study area relatively favorable conditions for water deer habitat prior to construction of the road. However, after the road construction the percentage over 0.6 dropped to 55.9% (decreasing about 7.8%), as a result of the habitat damage after construction. This study thus was able to show the habitat value quantitatively before and after the construction.

In addition, we compared the ecology at each of the 13 proposed eco-corridor locations, as well as the largest effect of construction on the net CHU increase at those locations to determine the most appropriate eco-corridor site. It is believed that the construction of mitigating eco-corridors is a cost-efficient method of providing the ecological passage of animals to other suitable habitat areas (Table 8). The total amount of CHU was 1211 prior to the construction and 299 after construction, which means that the habitat value was significantly reduced without a proper mitigation measure in the highway. However, due to the use of eco-corridors, it is possible to obtain a net increase of 363 (662–299) and it is possible to obtain improved habitat quality due to the eco-corridors, which allow the water deer to reach suitable habitat areas more effectively. Eco-corridor installation can be the best solution for increasing and/or maintaining habitat connectivity of rather large-sized mammals such as water deer in Korea as well as increasing habitat value and avoiding habitat fragmentation [28].

A comparison of CHU values of a proposed project through the HEP technique was conducted to quantitatively calculate the loss/increase of habitat suitable for water deer. This information can be applied to the existing environmental impact assessment framework to effectively improve the process through a quantitative assessment of environmental impacts from construction [29,30,31,32,33,34]. This study also showed that eco-corridor #4 can have a maximum net increase of 69.5 in terms of CHU value, thus indicating the most effective point to build the eco-corridor based on considerations of the CHU. Format et al. (2003) indicated several ways of improving habitat quality for various species in the wild, such as the use of fencing, or ditches for amphibians, but they pointed out that habitat connectivity should be the most important consideration and that cumulative effects must also be addressed to evaluate the extent of potential impacts. 

The HEP can be used to determine cost reductions for environmental protection by comparing the costs and effectiveness of proposed ecosystem restoration plans. In the United States, the HEP has been commonly used to make a reasonable choice in terms of the costs of environmental protection, including ecological effects, associated with business development projects. Considering the ecological and economic impacts of projects, studies on the mitigation of environmental impacts have been attempted such as SI and HSI models, and this technique can be applied to terrestrial and marine ecosystem [35,36,37]. To evaluate an ecosystem with uncertain characteristics and/or that has a variety of time and space characteristics, the following aspects are necessary. Research on the accumulation of objective data needed in the HEP modeling process and the establishment of SI, the basis for each species, needs to be conducted continuously.

There have been several recent studies related to the HEP process, such as a recent study (2018) on the SI mapping for the implementation of changes in order to protect the sand cat (*Felis margatia*), an endangered species [38], and a study on the suitable habitat selection for black bears (*Ursus americanus*) which combined HSI index analysis and GPS points collection in the North Carolina area [39]. In many countries, quantitatively assessing environmental habitats should become important. The process of evaluating an ecosystem and/or a natural resource is important, and further studies are needed to ensure sustainable environmental conservation in response to inevitable developments in the field of environmental impact assessment.

## 6. Conclusions

Ecological evaluation often fails to include quantitative evaluation as part of the existing environmental impact assessment (EIA) process. This study tried to quantify the long-term effects of a road construction project on an ecosystem using Habitat Evaluation Procedure (HEP) before and after the construction of a highway. The Korean water deer (*Hydropotes inermis*) was selected as the target species for study, and water deer habitat data were collected on vegetation cover, stream water density, geographic contour, land use class, and road networks. The Habitat Suitability Index (HSI) and CHU values for the water deer were estimated to investigate the major land cover classes, national river systems, and vegetation cover. Results showed that the CHU values became much higher with the creation of eco-corridors and minimized damage to habitat for water deer and this finding illustrated the importance of corridor installation to save habitat for water deer. A comparison was made of the 13 ecological mitigation corridors proposed for the road construction project area, and it was determined that the corridor that would result in the highest net increase in CHU (an increase of 69.5) was corridor #4, thus identifying it as the most appropriate corridor to properly connect water deer habitat. In sum, unlike other studies this study derived the net increase in quantitative values corresponding with different mitigation methods over time due to a road construction project and this procedure can be effectively utilized in the future to select the location of ecological corridors while considering the costs of constructing them. Continuous monitoring for the use of corridors and incidence of roadkill should be monitored to justify eco-corridor construction.

## Figures and Tables

**Figure 1 ijerph-16-01437-f001:**
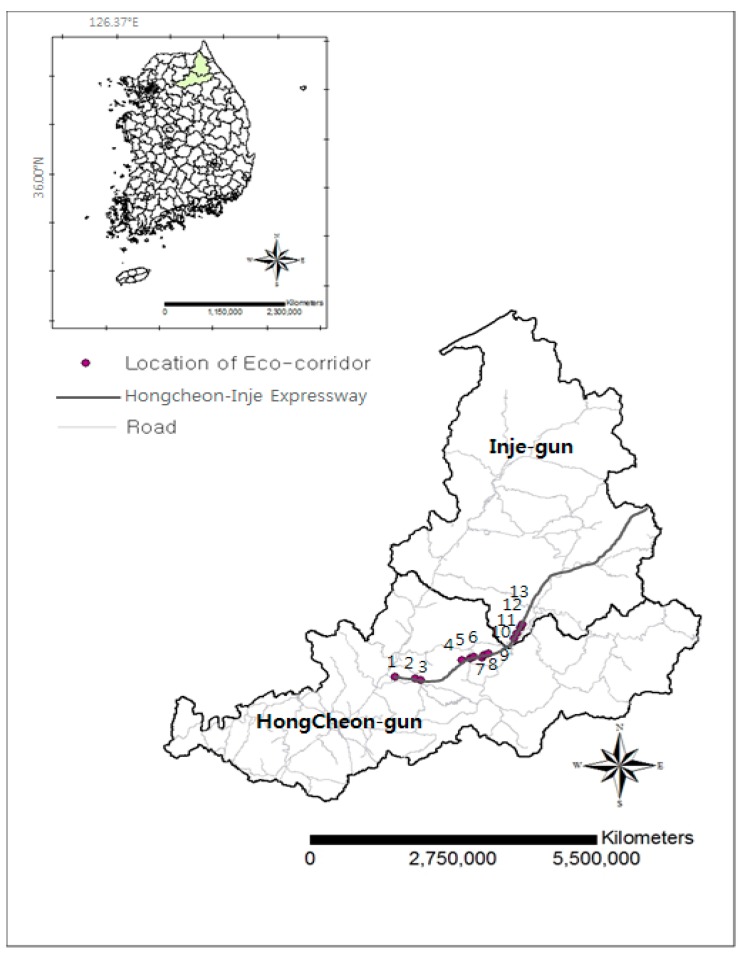
Hongcheon–Inje Express Highway in Gangwon Province, South Korea. Numbers in the map indicate the places where the eco-corridors were constructed.

**Figure 2 ijerph-16-01437-f002:**
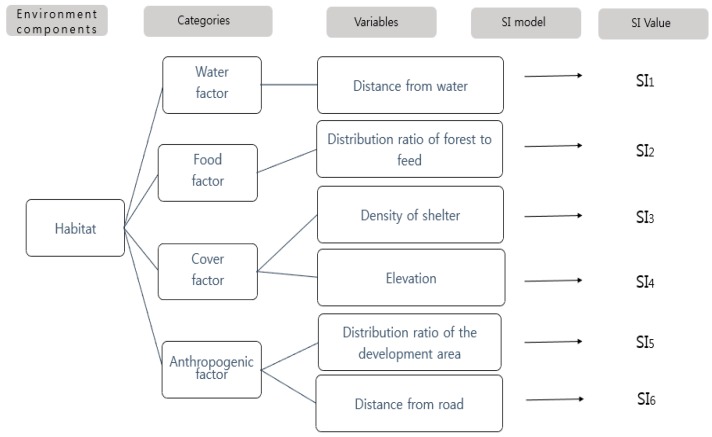
Suitability Index (SI) model selection and process used for Korean water deer (*Hydropotes inermis*).

**Figure 3 ijerph-16-01437-f003:**
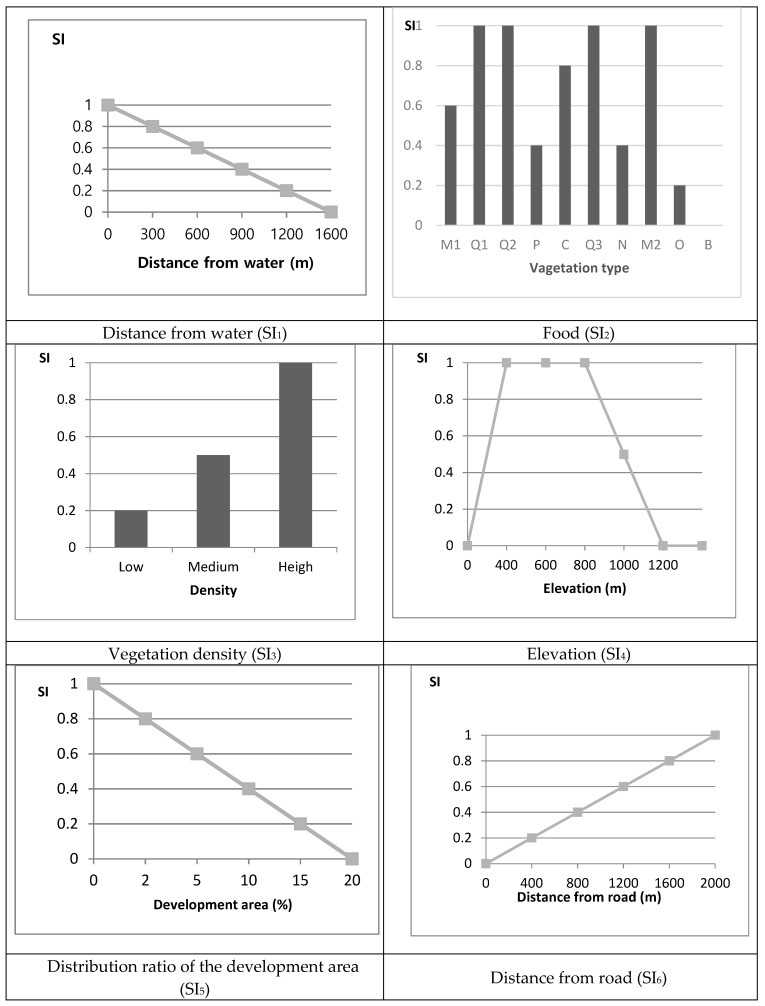
Suitability Index value of Korean water deer (*Hydropotes inermis*). Food (SI_2_) M1: Mixed oak forest; Q1: *Quercus variabilis*; Q2: *Q. dentate*; P: *Pinus* tree; C: *Carpinus laxiflora*; Q3: *Q. acutissima*; N: *Abies kora**iensis*; M2: *Q. mongolica*; O: Other tree; B: Base rock. SI_1–6_ were from Figure 2.

**Figure 4 ijerph-16-01437-f004:**
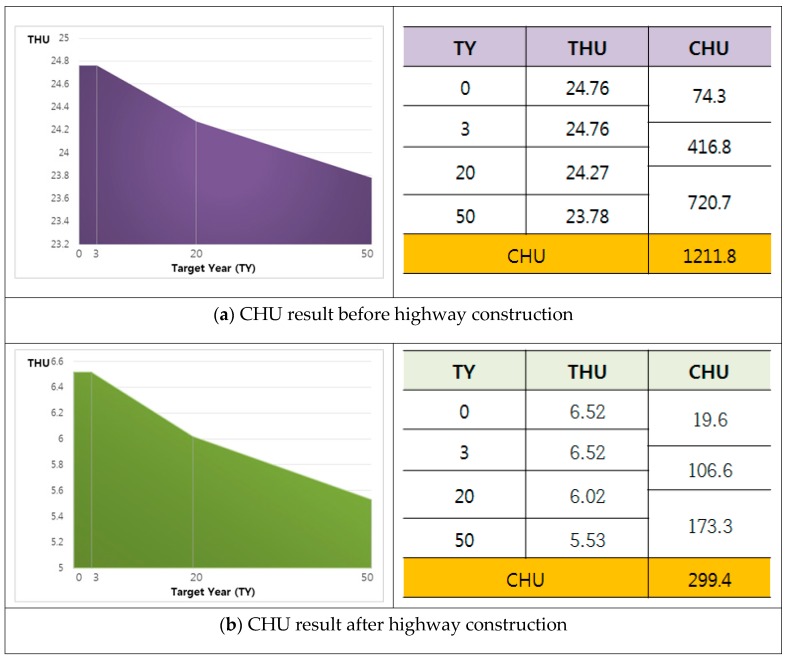
Cumulative Habitat Unit value (**a**) before highway construction, (**b**) after highway construction, and (**c**) after eco-corridor construction. TY: target year.

**Table 1 ijerph-16-01437-t001:** Analytical methods and index used by the Habitat Evaluation Procedure (HEP).

Index	Analytical Method
SI, Suitability Index	Value of 0–1 (highest) generated based on the ecological characteristics of the species.
HSI, Habitat Suitability Index	Value of 0–1 (highest) with the average SI for each species
HU, Habitat Unit	HSI multiplied by the areas HU=HSI×Total area
THU, Total Habitat Unit	Total of HU, which implies the quality and quantity of habitat type. THU=HUa+HUb+HUc a = 3, b = 20, c = 50 years after construction of eco-corridor
CHU, Cumulative Habitat Unit	THU accumulated over time showing the cumulative habitat unit of the species $CHU=\sum_{i=1}^{P}\left(\mathrm{AHSIi}\times \mathrm{Ai}\right)$ (*i* = year, *P* = time of duration, AHSI = weighted average of HSI, Ai: area of *i*th year)

**Table 2 ijerph-16-01437-t002:** Area and ratio of SI_1_ value in Hongcheon–Inje.

SI Value	Value	Area (km^2^)	Percentage (%)
0 ≤ SI < 0.2	1200–1600	350.56	16.76
0.2 ≤ SI < 0.4	900–1200	594.27	28.41
0.4 ≤ SI < 0.6	600–900	402.79	19.26
0.6 ≤ SI < 0.8	300–600	360.19	17.22
0.8 ≤ SI ≤ 1	0–300	383.82	18.35
**Total**		2091.63	100

**Table 3 ijerph-16-01437-t003:** Area and ratio of SI_2_ value in Hongcheon–Inje.

SI Value	Value	Area (km^2^)	%
SI = 0.6	Mixed oak forest	630.62	32.44
SI = 1.0	*Quercus variabilis*	73.60	3.79
SI = 1.0	*Quercus dentata*	0.54	0.03
SI = 0.4	Pine tree	299.50	15.41
SI = 0.8	*Carpinus laxiflora*	0.27	0.01
SI = 1.0	*Quercus acutissima*	0.94	0.05
SI = 0.4	Nut pine tree	0.40	0.03
SI = 1.0	Mongolian oak	929.78	47.83
SI = 0.2	Other	2.96	0.15
SI = 0.0	Base rock	5.37	0.28
**Total**		1943.97	100

**Table 4 ijerph-16-01437-t004:** Area and ratio of SI_3_ value in Hongcheon–Inje.

SI Value	Value	Area (km^2^)	%
SI = 0.2	Low	51.67	2.17
SI = 0.5	Medium	932.84	39.20
SI = 1.0	High	1395.21	58.63
**Total**		2379.72	100

**Table 5 ijerph-16-01437-t005:** Area and ratio of SI_4_ value in Hongcheon–Inje.

SI Value	Value	Area (km^2^)	%
0 ≤ SI < 1	0–400	932.11	6.19
SI = 1	400–600	1151.32	7.65
SI = 1	600–800	11,091.93	73.66
0.5 < SI ≤ 1	800–1000	993.30	6.60
0 < SI ≤ 0.5	1000–1200	586.64	3.90
SI = 0	<1200	303.20	2.01
**Total**		15,058.51	100

**Table 6 ijerph-16-01437-t006:** SI_5_ value in Hongcheon–Inje.

SI	Urban and Barren Area (%)
SI = 0.9	1.06

**Table 7 ijerph-16-01437-t007:** SI_6_ value in Hongcheon–Inje.

SI Value	Value	Area (km^2^)	%
0 ≤ SI < 0.2	0–400	785.72	17.17
0.2 ≤ SI < 0.4	400–800	378.72	8.28
0.4 ≤ SI < 0.6	800–1200	496.05	10.84
0.6 ≤ SI < 0.8	1200–1600	339.02	7.41
0.8 ≤ SI < 1.0	1600–2000	262.44	5.73
SI = 1.0	<2000	2314.44	50.57
**Total**		4576.39	100

**Table 8 ijerph-16-01437-t008:** Cumulative Habitat Unit, net increase result of 13 eco-corridor sites.

**Eco-Corridor Sites**	**1**	**2**	**3**	**4**	**5**	**6**	**7**
No construction	90.89	72.48	78.36	129.39	119.58	107.80	103.88
After construction	18.27	22.43	23.41	25.38	39.11	22.43	23.02
After eco-corridor	67.33	65.46	65.07	94.90	77.24	91.57	61.15
Net increase	49.07	44.03	41.66	69.53	38.13	69.13	38.13
**Eco-corridor sites**	**8**	**9**	**10**	**11**	**12**	**13**	**Total**
No construction	99.95	107.80	74.44	88.18	88.18	88.18	**1211**
After construction	23.02	23.41	24.98	21.45	22.24	24.45	**299**
After eco-corridor	61.15	61.54	66.45	56.91	66.25	69.39	**662**
Net increase	38.13	38.13	41.48	35.46	44.01	44.94	**363**

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
