# Peer review of "Application of Habitat Evaluation Procedure with Quantifying the Eco-Corridor in the Process of Environmental Impact Assessment"

_ijerph, 2019, doi:10.3390/ijerph16081437_

Round 1

Reviewer 1 Report

The paper presents very interesting and valuable method, but needs some general improvement in structure and minor adjustment in text to be ready for publishing.

General remarks:

 The abstract is clear and engage readers in the matter of the text, maybe it could be a bit shortened.

Key words could include couple of words about methods and place as well.

The introduction provides all expected information, only the Habitat Evaluation Procedure needs more information here. You cannot require reader to wait for section of Results, where there is broader explanation of the method.

Material and methods are presented very well, excluding some particular details, I described in “Detailed remarks”

Results are clear, full of interesting charts and schemes, but discussion needs general reconstruction, being mostly sumarising results, not real discussion, which have to refer results in comparison with another researchers/authors. There is lack of conclusions in the paper, some of them I can see in the “discussion” section, but I strongly recommend add in the separate section 6. called Conclusions.

Concluding, in my opinion text is very good and valuable, but lacking of conclusions and wrong discussion forces Authors to improve it.

Detailed remarks:

Line 83 Figure 1 – what are numbers on the map? I believe there are ecological routes? Please add key (legend) below the map.

Line 105 What do You mean by the term “geographic contour lines” – it is delimitation of the area researched?

Line 136 Figure 3 – Where the criteria presented in charts came from? Please supply the caption.

Line 93 where is the appendix You are writing about?

Author Response

The paper presents very interesting and valuable method, but needs some general improvement in structure and minor adjustment in text to be ready for publishing.

General remarks:

 The abstract is clear and engage readers in the matter of the text, maybe it could be a bit shortened.

Key words could include couple of words about methods and place as well.

habitat evaluation procedure; ecological evaluation; road construction; roadkill; eco-corridor => changed habitat evaluation procedure => Gangwon province

The introduction provides all expected information, only the Habitat Evaluation Procedure needs more information here. You cannot require reader to wait for section of Results, where there is broader explanation of the method.

The HEP involves assessment based on accounting techniques in which the habitat of the targeted species can be quantified, and the 'quality' of the species’ habitat is the HSI value generated by calculating areas that conform to the target species’ habitat within its distribution => moved to introduction

Material and methods are presented very well, excluding some particular details, I described in “Detailed remarks”

             => Thank you

Results are clear, full of interesting charts and schemes, but discussion needs general reconstruction, being mostly sumarising results, not real discussion, which have to refer results in comparison with another researchers/authors. There is lack of conclusions in the paper, some of them I can see in the “discussion” section, but I strongly recommend add in the separate section 6. called Conclusions.

             => Conclusions were added and included important finding there

Concluding, in my opinion text is very good and valuable, but lacking of conclusions and wrong discussion forces Authors to improve it.

 Detailed remarks:

Line 83 Figure 1 – what are numbers on the map? I believe there are ecological routes? Please add key (legend) below the map.

Added ‘Numbers in the map indicating the places where the eco-corridor was constructed.’

Line 105 What do You mean by the term “geographic contour lines” – it is delimitation of the area researched?

geographic contour lines => contour lines

Line 136 Figure 3 – Where the criteria presented in charts came from? Please supply the caption.

SI1~6 were from Figure 2

Line 93 where is the appendix You are writing about?

Added appendix 1. Thank you.

Reviewer 2 Report

The paper “Application of Habitat Evaluation Procedure for maximizing habitat value for road construction in the process of environmental impact assessment” is a Korean case study that focuses on the evaluation of the environmental impacts caused by road construction. This is an interesting topic that fit the aims of the journal. Although I am not a native speaker, I suggest an English language revision. In my opinion, the paper can be published after several changes. Therefore, my judgment is MAJOR REVISIONS.

I think that the title must be changed because the issue of the paper concerns a specific case study and, for this, it cannot be generalized.

Abstract:

Line 25: Why the recovery value after 50 years is 549.6? It is not specified in any part of the text.

Line 29: the authors affirms that “the study derived the net profit quantitative values corresponding with different mitigation methods over time due applied to a road construction project” but any mitigation method is specified in the text (e.g. Discussion section).

The authors should specify also in this section the research question of the paper (lines 68-70).

Introduction:

The research that is illustrated is part of the theme of Road Ecology, a scientific sector for some years in great international success. I suggest to the authors to extend this part, the following references could be considered:

Fahrig L (2003) Effects of habitat fragmentation on biodiversity. Ann Rev Ecol Syst 34:487–515. https ://doi.org/10.1146/annurev.ecols ys.34.01180 2.13241 9

Forman RTT, Sperling D, Bissonette JA, Clevenger AP, Cutshall CD, Dale VH, Fahrig L, France RL, Goldman CR, Heanue K (2002) Road ecology: science and solutions. Island Press, Washington (ISBN 9781559639330).

Materials and Methods:

The authors should describe the eco-corridor principal characteristics.  

Figure 1: Insert a base map layer (e.g. Google Earth recent images). Furthermore, the Eco-corridor can be represented with a lines geometry. Why the authors used the points geometry? Are they wildlife crossings? It is not clear.

Data collection and analysis: The authors should better specify all data technical characteristics and their data sources.

Theory/calculations:

Table 1: What are HUa, HUb and HUc? It is not specified.

What is the statistical unit used to calculate the indexes? The authors should well describe it. The authors should insert this description in the Methods section.

Results:

Table 3: Food not Value;

Figure 4: The authors could realize only a figure that reports all 3 Cumulative Habitat Unit values (a,b and c) in such a way to use an identical scale values.

The authors should better discuss on the net loss value due the road construction (CHU is 25% of the initial value). Furthermore, after eco-corridor construction, the CHU value is 50% of the initial value. This is very important and should be given a more detailed in the discussion section.

- Discussion:

The whole discussion should be rephrased. First part just repeats some facts stated already below. State only, what is new, how you contributed to science and if there can anything else evolve from your findings.The authors should add a conclusions section.     

Author Response

I think that the title must be changed because the issue of the paper concerns a specific case study and, for this, it cannot be generalized.

Changed the title => Application of Habitat Evaluation Procedure with quantifying the eco-corridor in the process of Environmental Impact Assessment

Abstract:

Line 25: Why the recovery value after 50 years is 549.6? It is not specified in any part of the text.

Changed to ‘Results showed that the environmental impact in the road construction project area would result in a net ecological loss value of 1,211 without installation of eco-corridor, and reduced to 662 with eco-corridor, which is a 549 increase in the net value after 50 years of the mitigation plan.’

Line 29: the authors affirms that “the study derived the net profit quantitative values corresponding with different mitigation methods over time due applied to a road construction project” but any mitigation method is specified in the text (e.g. Discussion section).

Changed sentence to make it clearer

The authors should specify also in this section the research question of the paper (lines 68-70).

Changed to ‘The purpose of this research was to construct a net ecological value of eco-corridor before and after the road construction based on the HEP technique as a way to explore the method as an effective quantitative assessment supplement to the current EIA framework.’

Introduction:

The research that is illustrated is part of the theme of Road Ecology, a scientific sector for some years in great international success. I suggest to the authors to extend this part, the following references could be considered:

included both citations. Thank you

28. Fahrig L (2003) Effects of habitat fragmentation on biodiversity. Ann Rev Ecol Syst 34:487–515. https ://doi.org/10.1146/annurev.ecols ys.34.01180 2.13241 9

34. Forman RTT, Sperling D, Bissonette JA, Clevenger AP, Cutshall CD, Dale VH, Fahrig L, France RL, Goldman CR, Heanue K (2002) Road ecology: science and solutions. Island Press, Washington (ISBN 9781559639330).

Materials and Methods:

The authors should describe the eco-corridor principal characteristics.  

Added ‘Eco-corridor was designed with 40 m in width and 200-700 m in length depending on shape of mountain contour lines and usually the top part was planted with vegetation of species found around the areas to minimized negative impact for the use of structures for the animals.’

Figure 1: Insert a base map layer (e.g. Google Earth recent images). Furthermore, the Eco-corridor can be represented with a lines geometry. Why the authors used the points geometry? Are they wildlife crossings? It is not clear.

Eco-corridor usually constructed in the line vertical to roads so that in GIS this should be viewed as points

Data collection and analysis: The authors should better specify all data technical characteristics and their data sources.

The roadkill data were gathered by experts in the Korean Express Highway and they visited the highways in the early morning so that identifying species was possible.

Theory/calculations:

Table 1: What are HUa, HUb and HUc? It is not specified.

a=3, b=20, c=50 years after construction of eco-corridor

What is the statistical unit used to calculate the indexes? The authors should well describe it. The authors should insert this description in the Methods section.

The value was calculated for a comparison with the eco-corridor effects in terms of habitat unit.

Results:

Table 3: Food not Value;

Each species was given SI value

Figure 4: The authors could realize only a figure that reports all 3 Cumulative Habitat Unit values (a,b and c) in such a way to use an identical scale values.

We used the same scale for each category and value was generated

The authors should better discuss on the net loss value due the road construction (CHU is 25% of the initial value). Furthermore, after eco-corridor construction, the CHU value is 50% of the initial value. This is very important and should be given a more detailed in the discussion section.

We discussed this in more detail being clearer and indicated this is an important findings for this study.

- Discussion:

The whole discussion should be rephrased. First part just repeats some facts stated already below. State only, what is new, how you contributed to science and if there can anything else evolve from your findings.The authors should add a conclusions section.     

Results showed that the CHU became much higher with eco-corridor and minimizing damage to habitat for waterdeer providing 55% (662/1211) when compared with no construction of corridors and this finding illustrated the importance of corridor installation to save habitat.

Reviewer 3 Report

While the authors have demonstrated the use of HEP to quantitatively assess habitat value using CHU, I believe this paper still requires additional work and editing before publication can be considered.  Given the paper’s technical approach, the main shortcoming is the lack of clear and in-depth description of their methodology, which I will list below:

1. Spatial Analysis (lines 103-114): exactly what kind of spatial analysis was used to derive the SI values.  It unclear if the authors used vector overlay analysis or raster calculation or a combination of both.  If raster calculation, what is the resolution of the raster grid cell?  This should be elaborated for each SI.

2. Line 115, what “statistical analyses” were performed and for what purposes?

3. Line 122, description in Table 1 should be consistent with the text in the manuscript.  For example, HSI is described in the table as “…totaling of SI for each species” but shown as the average in line 150.  For CHU, it’s described as “THU multiplied by time…” which does not agree with the expression in the table.

4. Line 152-153 stated that “The Habitat Unit (HU) value comprises the HSI value obtained for all study stages in each region multiplied by the HU values from each area of study.”  This is confusion and it not clear exactly how HU values are determined.

5. Eco-corridor areas: the authors indicated that there are 13 proposed eco-corridor sites.  How similar or different these proposed sites are in terms of what’s proposed?  Without knowing the specifics of the 13 eco-corridor sites and their surrounding environment, it’s very difficult to make sense of the results presented in Table 8 (line 218) as to why their CHU values vary.

6. Figure 4 (line 203) presents the three scenarios which are fine but it’s unclear where the TY 3 came from and included?  Why the THU remains the same between TY 0 and 3?  Why the CHU values decline after TY 3 across all three scenarios?  Why the slope of the CHU plot line change after TY 20?  It would be helpful if the authors can provide further information to the above questions in the paper.

Author Response

1. Spatial Analysis (lines 103-114): exactly what kind of spatial analysis was used to derive the SI values.  It unclear if the authors used vector overlay analysis or raster calculation or a combination of both.  If raster calculation, what is the resolution of the raster grid cell?  This should be elaborated for each SI.

   3.1.1.1. Distance from water (SI1)

 Waterdeer prefer areas with water so that GIS Euclidean Distance measure with Spatial Analyst tool was applied with a maximum distance of 1.6 km and value averaged 0.40(s.e=±0.03) covering 35.57% over 0.6.

3.1.1.2. Distribution ratio of forest to feed (Food) (SI2)

Food resources of waterdeer using vegetation around the region averaging 0.76(s.e=±0.03) and 84.15% over SI2 value greater than 0.6. Mongolian oaks (Quercus mongolica) covering 47.84% and mixed oak forests were 32.44% indicating areas were suitable habitat for waterdeer

3.1.1.3. Density of shelter (Vegetation) (SI3)

High density of vegetation in the region was estimated averaging 0.79(s.e=±0.03), and maximum coverage was 58.63% and medium averaged 39.20% (>0.5) so that most areas were suitable for waterdeer habitat.

3.1.1.4. Elevation (SI4)

Elevation was calculated with contour map averaging 0.94(s.e=±0.05) with 400-800m (81.31%) indicating presumably most areas being suitable for waterdeer habitat.

3.1.1.5. Distribution ratio of the development areas (SI5)

Land coverage map was applied showing forest areas (91.61%) and urban areas (0.98%) averaging 0.9 value of SI.

3.1.1.6. Distance from road (SI6)

With similar method to SI1 the Euclidean Distance was estimated averaging 0.69(s.e=±0.04), and the most areas were 1.0 (50.57%) and greater than 0.6 (63.71%) indicating areas were far from the road.

2. Line 115, what “statistical analyses” were performed and for what purposes?

Sentence corrected => ‘Map production and space analysis with Arc View and GIS was done [18-20].’

3. Line 122, description in Table 1 should be consistent with the text in the manuscript.  For example, HSI is described in the table as “…totaling of SI for each species” but shown as the average in line 150.  For CHU, it’s described as “THU multiplied by time…” which does not agree with the expression in the table.

Corrected, => HSI used the average of all SI values

Corrected, => accumulated with time

4. Line 152-153 stated that “The Habitat Unit (HU) value comprises the HSI value obtained for all study stages in each region multiplied by the HU values from each area of study.”  This is confusion and it not clear exactly how HU values are determined.

Corrected,=> The Habitat Unit (HU) value was generated with HSI value obtained for all study stages in each region multiplied by the HU values from each area of study.

5. Eco-corridor areas: the authors indicated that there are 13 proposed eco-corridor sites.  How similar or different these proposed sites are in terms of what’s proposed?  Without knowing the specifics of the 13 eco-corridor sites and their surrounding environment, it’s very difficult to make sense of the results presented in Table 8 (line 218) as to why their CHU values vary.

The differences come from the each corridor with land coverage map, and even though the structure was made in the highway the cumulative habitat unit value was not consistent.

6. Figure 4 (line 203) presents the three scenarios which are fine but it’s unclear where the TY 3 came from and included?  Why the THU remains the same between TY 0 and 3?  Why the CHU values decline after TY 3 across all three scenarios?  Why the slope of the CHU plot line change after TY 20?  It would be helpful if the authors can provide further information to the above questions in the paper.

TY 3 was included to see the changes after highway construction so that there was not so much changes.

CHU with time can be changing and CHU before construction was not changing so much (y-scale); however, CHU after construction was declining rapidly and CHU after corridor the value was increasing. The figure illustrated the total amount of CHU so it looks decreasing due to heavy change after construction.

Round 2

Reviewer 1 Report

The paper presents very interesting and valuable method, now it is improved and now needs only slight adjustment to be ready for publishing.

The abstract is clear and engage readers in the matter of the text. Additional explanations improved the text. Key words was changed and now include all proper terms.

The introduction provides all expected information, authors moved the explanation about Habitat Evaluation Procedure method from methods to the introduction, making it more informative.

Material and methods are presented very well, but I still believe that resignation from that “short” subsections like 3.1.1.1., 3.1.1.2. etc. make the text better and more “readable”. Maybe only one subsection, 3.1.1. and then bullets will be enough?

Results are clear, and well done.

Section of discussion was significantly improved, a couple of interesting references were added, then now it looks better.

Section of Conclusions was added as a separate part of the text. Authors move some text from the former discussion to the new section. Conclusions are maybe to laconic, but provide all information and statements needed.

Concluding, text is much better than previously and I can recommend it for publication after suggested adjustment.

Author Response

The paper presents very interesting and valuable method, now it is improved and now needs only slight adjustment to be ready for publishing.

The abstract is clear and engage readers in the matter of the text. Additional explanations improved the text. Key words was changed and now include all proper terms.

The introduction provides all expected information, authors moved the explanation about Habitat Evaluation Procedure method from methods to the introduction, making it more informative.

=> Thank you.

Material and methods are presented very well, but I still believe that resignation from that “short” subsections like 3.1.1.1., 3.1.1.2. etc. make the text better and more “readable”. Maybe only one subsection, 3.1.1. and then bullets will be enough?

=> The subsection changed to bullets as reviewer suggested

Results are clear, and well done.

Section of discussion was significantly improved, a couple of interesting references were added, then now it looks better.

Section of Conclusions was added as a separate part of the text. Authors move some text from the former discussion to the new section. Conclusions are maybe to laconic, but provide all information and statements needed.

Concluding, text is much better than previously and I can recommend it for publication after suggested adjustment.

=> Thank you

Reviewer 2 Report

All my required changes have been made and the manuscript has been significantly improved. I think that  the paper can be published in IJERPH after a english revision by a native speaker.

Author Response

All my required changes have been made and the manuscript has been significantly improved. I think that  the paper can be published in IJERPH after a english revision by a native speaker.

=> Thank you. The manusctipt was checked by native english speaker.

Reviewer 3 Report

Thanks to the authors for addressing my previous comments. The revised version is much improved and the authors have made substantial revisions. I think the manuscript still requires professional English editing before publication.

Author Response

Thanks to the authors for addressing my previous comments. The revised version is much improved and the authors have made substantial revisions. I think the manuscript still requires professional English editing before publication.

=> Thank you. The manuscript was checked by English native.